# Influenza A viruses are transmitted via the air from the nasal respiratory epithelium of ferrets

Mathilde Richard [1], Judith M.A. van den Brand [1], Theo M. Bestebroer[1], Pascal Lexmond[1], Dennis de Meulder[1], Ron A.M. Fouchier[1], Anice C. Lowen [2,3] & Sander Herfst [1]*

Human influenza A viruses are known to be transmitted via the air from person to person. It is unknown from which anatomical site of the respiratory tract influenza A virus transmission occurs. Here, pairs of genetically tagged and untagged influenza A/H1N1, A/H3N2 and A/H5N1 viruses that are transmissible via the air are used to co-infect donor ferrets via the intranasal and intratracheal routes to cause an upper and lower respiratory tract infection, respectively. In all transmission cases, we observe that the viruses in the recipient ferrets are of the same genotype as the viruses inoculated intranasally, demonstrating that they are expelled from the upper respiratory tract of ferrets rather than from trachea or the lower airways. Moreover, influenza A viruses that are transmissible via the air preferentially infect ferret and human nasal respiratory epithelium. These results indicate that virus replication in the upper respiratory tract, the nasal respiratory epithelium in particular, of donors is a driver for transmission of influenza A viruses via the air.

[1] Department of Viroscience, Erasmus MC University Medical Center, Center for Research on Influenza Pathogenesis (CRIP) Center of Excellence for Influenza Research and Surveillance (CEIRS), Rotterdam, the Netherlands. [2] Department of Microbiology and Immunology, Emory University School of Medicine, Atlanta, GA 30322, USA. [3] Emory-UGA Center of Excellence for Influenza Research and Surveillance (CEIRS), Atlanta, GA 30322, USA. *email: s.herfst@erasmusmc.nl

Millions of people have lost their lives due to influenza A virus (IAV) epidemics and pandemics. Prevention and control of IAV infections are based on vaccination and treatment. However, a better fundamental understanding of IAV transmission could help to design additional appropriate intervention strategies, especially in health care settings. Transmission of IAVs between humans can occur via direct or indirect person-to-person contact and through the air via respiratory droplets or aerosols. The relative contribution of each route of transmission is still under debate[1–3], yet it is widely accepted that aerosol or respiratory droplet transmission is the key factor for the rapid spread and continued circulation of IAVs in humans. Evidence for this transmission route is supported by the direct detection of influenza virus genomes and viable influenza virus particles in aerosols and/or respiratory droplets from breathing[4–7], sneezing or coughing individuals[6,8–11], and in the air in hospitals and healthcare settings[12–15]. Moreover, transmission of IAVs via the air is also supported by animal studies in ferrets[16–18] and guinea pigs[19,20]. In the ferret transmission model, pandemic and seasonal IAVs isolated from humans are transmitted from an infected donor ferret to an exposed recipient ferret via the air, whereas avian influenza viruses are generally not[21]. Our ferret transmission model does not allow distinction between transmission via aerosols or respiratory droplets; therefore, the terminology "airborne" transmission will be used in the rest of this manuscript when referring to aerosols and/or respiratory droplet transmission.

The ferret transmission model has also been used to investigate the viral properties that are necessary for airborne transmission of IAVs[21–25]. Among them is the binding of the hemagglutinin (HA), one of the virus surface glycoproteins, to sialic acids (SA) receptors linked to the penultimate sugar, galactose, by a α2,6 linkage (α2,6-SA)[21–25]. In general, the HA of human(-adapted) IAVs preferentially recognizes α2,6-SA, whereas that of avian IAVs preferentially binds to α2,3-SA[26,27]. In humans, α2,6-SA receptors are predominantly present on ciliated cells in the upper respiratory tract (URT), i.e., in the nasal turbinates, paranasal sinuses, pharynx and larynx, and in the upper part of the lower respiratory tract (LRT), i.e., in the trachea and bronchus[28,29]. In contrast, α2,3-SA receptors are mainly present on bronchiolar non-ciliated cuboidal cells and alveolar type II pneumocytes of the LRT[28,29]. However, it still remains unknown from which exact anatomical site of the respiratory tract airborne transmissible IAVs are expelled and whether α−2,6-SA preference is critical at the donor or recipient level, or both. To determine whether airborne transmissible IAVs originate from tissues of the URT or LRT, we here investigate the transmission of untagged and genetically tagged variant IAVs[30,31] upon simultaneous inoculation at a different anatomical site: the URT or LRT of ferrets. Our results demonstrate that airborne transmissible IAVs are transmitted from the URT, more specifically from the nasal turbinates of ferrets, which correlates with high infectivity in the ferret and human nasal respiratory epithelium in vivo and in vitro, respectively. Finally, the tropism for nasal respiratory epithelium in ferrets is shown to be determined by mammalian adaptation markers in the HA protein.

## Results

**The site of generation of airborne IAVs is the URT**. In order to understand from which anatomical site of the respiratory tract airborne transmissible IAVs are expelled, untagged and genetically tagged variant (var) versions of the A/Netherlands/602/2009 virus (A/H1N1 and A/H1N1$_{var}$)[30] were used to inoculate donor animals in the ferret transmission model[18,32]. The A/H1N1$_{var}$ virus carries a single synonymous substitution

per gene segment, so that it can be differentiated from the A/H1N1 virus. Two ferrets (Donor 1 and 2) were inoculated intranasally with the A/H1N1 virus and intratracheally with the A/H1N1$_{var}$ virus. Two additional ferrets were inoculated with the opposite placement of viruses (Donor 3 and 4), in order to correct for potential small differences in transmissibility as a result of the introduced substitutions in the var virus. Four hours after inoculation (hpi) of donor ferrets, recipient ferrets were placed in an opposite cage separated by two steel grids, 10 cm apart, to avoid direct contact transmission. Throat and nose swabs were collected from donor and recipient ferrets every 24 and 12 h, respectively. The anatomical sites targeted by the different inoculation routes and sampling methods are shown schematically in Supplementary Fig. 1.

We have previously shown that double inoculations at different anatomical sites of the respiratory tract of ferrets led to compartmentalization of the viruses, which greatly restricted reassortment[30]. Therefore, next-generation sequencing was performed on only one gene, the PB2 gene, to determine whether the throat and nose swabs collected from the donor and recipient ferrets contained the A/H1N1, A/H1N1$_{var}$ viruses or a mix of both viruses. For donor ferrets, swabs collected at 1-day post-inoculation (dpi), at the day that transmission to recipient ferrets was observed and at the last day that donor ferrets were virus positive (as determined by virus titration) were processed for next-generation sequencing. For recipient ferrets, the first and the last swabs that were positive (threshold value in RT-qPCR (Ct value) < 35) were subjected to next-generation sequencing. Virus inocula were also processed for next-generation sequencing to confirm clonality (see the "Methods" section).

In the nose swabs of donor ferrets, viruses were of the same genotype as the virus that was inoculated in the nose (Fig. 1). However, in the throat swabs of three out of four donor ferrets, a low amount of the virus that was instilled in the LRT was also detected (Donor 1 (day 1 (17.8%), day 4 (76.5%), day 6 (22.2%)), donor 3 (day 1 (2.4%)) and donor 4 (day 1 (94.6%), day 5 (9.9%) and day 6 (20.8%)). Airborne transmission occurred between all four pairs of ferrets. Each time, the predominant viruses detected in recipient ferrets were of the same genotype as the virus that was inoculated intranasally in donor ferrets. Only very low proportions of the A/H1N1$_{var}$ virus, which was inoculated in the LRT, was detected in the throat swabs of Recipient 1 at 4 dpi (2.6%) and at 7 dpi (2%). The virus that was used to inoculate ferrets intratracheally was only detected in throat swabs collected from three donor ferrets (donor 1, 3, and 4). Therefore, to allow confirmation that the virus instilled intratracheally was actively replicating in the LRT, a similar experiment was conducted, and this time the donor ferrets were euthanized when transmission to recipient ferrets was demonstrated, or at the latest at 5 dpi (donor/recipient pairs 5–8, Fig. 2 and in more detail in Supplementary Fig. 2). Tissues of the different parts of the respiratory tract (nasal turbinates, trachea (upper and lower part), lung (left and right lobes)) were harvested and homogenized, and subsequently subjected to virus titration and next-generation sequencing. Furthermore, in order to study whether transmission from the upper respiratory tract of donor ferrets would also be observed with other airborne transmissible IAVs, similar experiments were conducted using untagged and genetically tagged (var) pairs of a human A/H3N2 virus (A/H3N2 and A/H3N2$_{var}$; A/Panama/2007/99; donor/recipients pairs 9–12; Fig. 2 and in more detail in Supplementary Fig. 3)[33] and of an airborne transmissible highly pathogenic avian influenza A/H5N1 virus (A/H5N1$_{AT}$ and A/H5N1$_{AT-var}$; A/Indonesia/5/2005 carrying nine substitutions, see the "Methods section"; donor/recipient pairs 13–16; Fig. 2 and in more detail in Supplementary Fig. 4)[32]. The predominant virus genotypes detected in the nasal turbinates

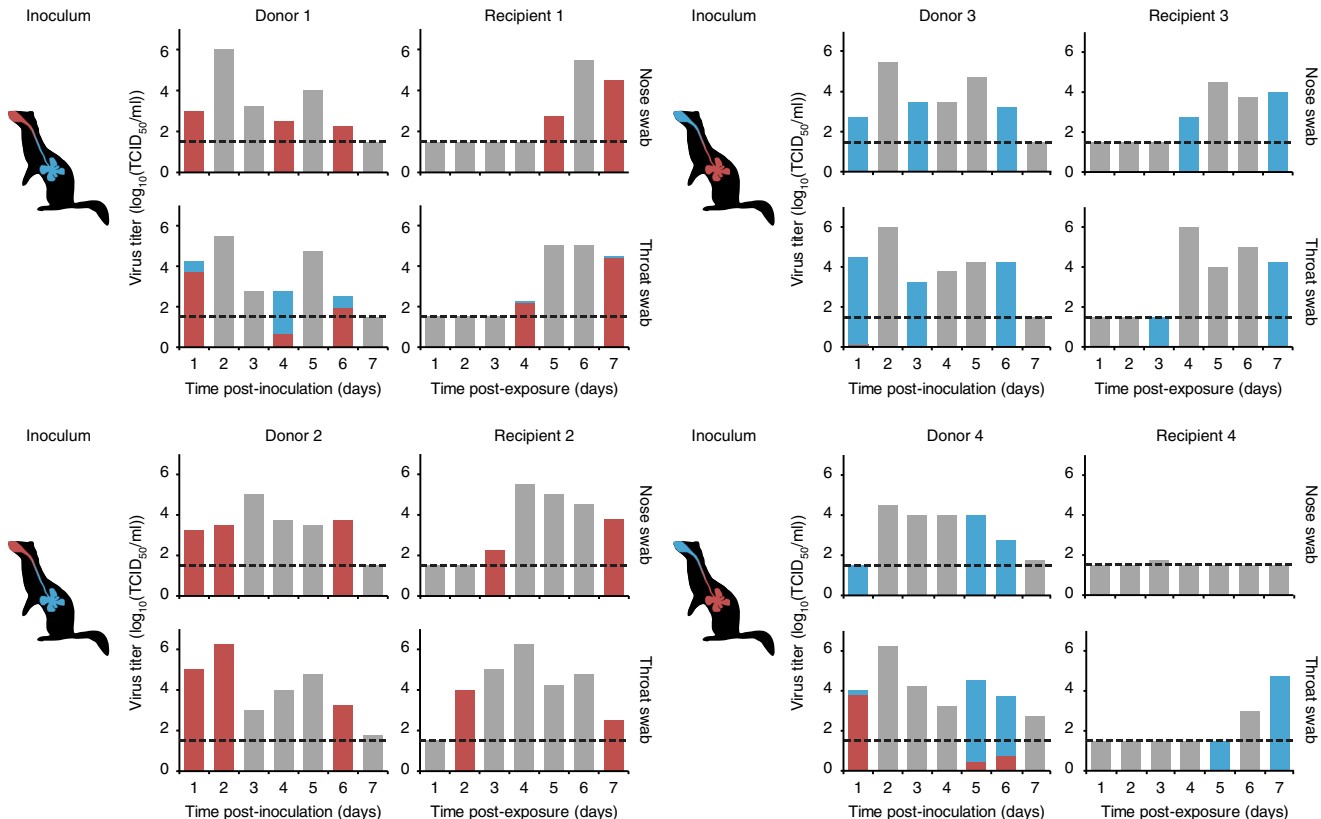

**Fig. 1 The A/H1N1 virus was transmitted from the upper respiratory tract of ferrets.** Donor ferrets 1 and 2 were inoculated intranasally with $10^5$ $TCID_{50}$ of the A/H1N1 virus (shown in red) and intratracheally with $10^5$ $TCID_{50}$ of the A/H1N1$_{var}$ virus (shown in blue). Donor ferrets 3 and 4 were inoculated with the opposite placement of viruses. Recipient ferrets were added to the adjacent cage at 4 hpi. Virus titers in the nose and throat swabs were determined by $TCID_{50}$ assay and are indicated on the y-axis. The limit of detection of the virus titration is shown by the dotted line. The proportions of untagged (red) and tagged (blue) viruses, as determined by next-generation sequencing on the PB2 gene, are indicated by the colored bars. The gray bars correspond to samples that were not included in the next-generation sequencing. Source data are provided as a Source Data file.

of all donor ferrets were the same as those detected in the nasal swabs. In the throat swabs of all donor ferrets, mixtures of untagged and tagged viruses were present, as observed in the first experiment with the A/H1N1 virus. In the trachea and lung samples, the predominant virus genotype was that of the virus that was inoculated intratracheally, with the exception of donor 10 (A/H3N2) and 15 (A/H5N1$_{AT}$). However, in both cases, the virus inoculated intratracheally was also detected, confirming that the virus instilled in the LRT was replicating. Titers in the LRT of donor ferrets inoculated with the A/H3N2 virus were lower than those in ferrets inoculated with A/H1N1 or A/H5N1 viruses. However, viral RNA was amplified and sequenced from each sample. A/H1N1, A/H3N2, and A/H5N1$_{AT}$ viruses were transmitted via the airborne route between two out of four, three out of four and two out of four donor/recipient pairs, respectively. Transmission was defined here by the detection of two consecutive swabs with a threshold value in RT-qPCR (Ct value) < 35. Despite the fact that infectious virus titers were detected only in one throat swab of Recipient 13 (Supplementary Fig. 4), viral RNA was amplified from the other swabs of Recipient 13 and Recipient 15, allowing the characterization of the nature of the virus that had transmitted. On all occasions when airborne transmission was observed, the virus that was inoculated intranasally in the donor ferrets was detected in the swabs of recipient ferrets. Despite detection of a very low proportion of the virus that was inoculated intratracheally in the donors in a minority of recipients (Recipients 7, 9, and 13), airborne transmissible IAVs were mainly transmitted from the

upper respiratory tract and not from the trachea and lungs of donor ferrets.

**Airborne IAVs infect the ferret nasal respiratory epithelium.** Next, the cell tropism in the URT that promotes airborne transmission was investigated. The URT is composed of the nasal turbinates (which comprises the nasal respiratory epithelium and the olfactory epithelium), paranasal sinuses, pharynx and the larynx. In a previous study, it was observed that human IAVs (A/H1N1 A/Netherlands/602/2009 and A/H3N2 A/Netherlands/177/2008), in contrast to the avian A/H5N1 virus (A/Indonesia/5/2005), abundantly infected the nasal respiratory epithelium of ferrets, located in the rostral part of the nasal turbinates, upon intranasal inoculation[34]. In ferrets inoculated with A/H1N1 and A/H3N2 viruses, the nasal respiratory epithelium was infected as early as 1 dpi, the lining epithelium was damaged because of the infection and start of regeneration of the epithelium was observed at 3 dpi. In order to understand whether this nasal respiratory tropism was a common trait of airborne-transmissible influenza viruses, immunohistochemical stainings were performed on nasal turbinates, collected at 2 dpi, of three ferrets inoculated with either A/H5N1$_{AT}$ or A/H5N1, as a control. The time point 2 dpi was chosen based on the observation that regeneration of the epithelium was observed as early as 3 dpi in ferrets inoculated with A/H1N1 and A/H3N2 viruses. The percentage of epithelium that was antigen-positive in the nasal respiratory and olfactory epithelia, as well as the infectious virus titers in the homogenized

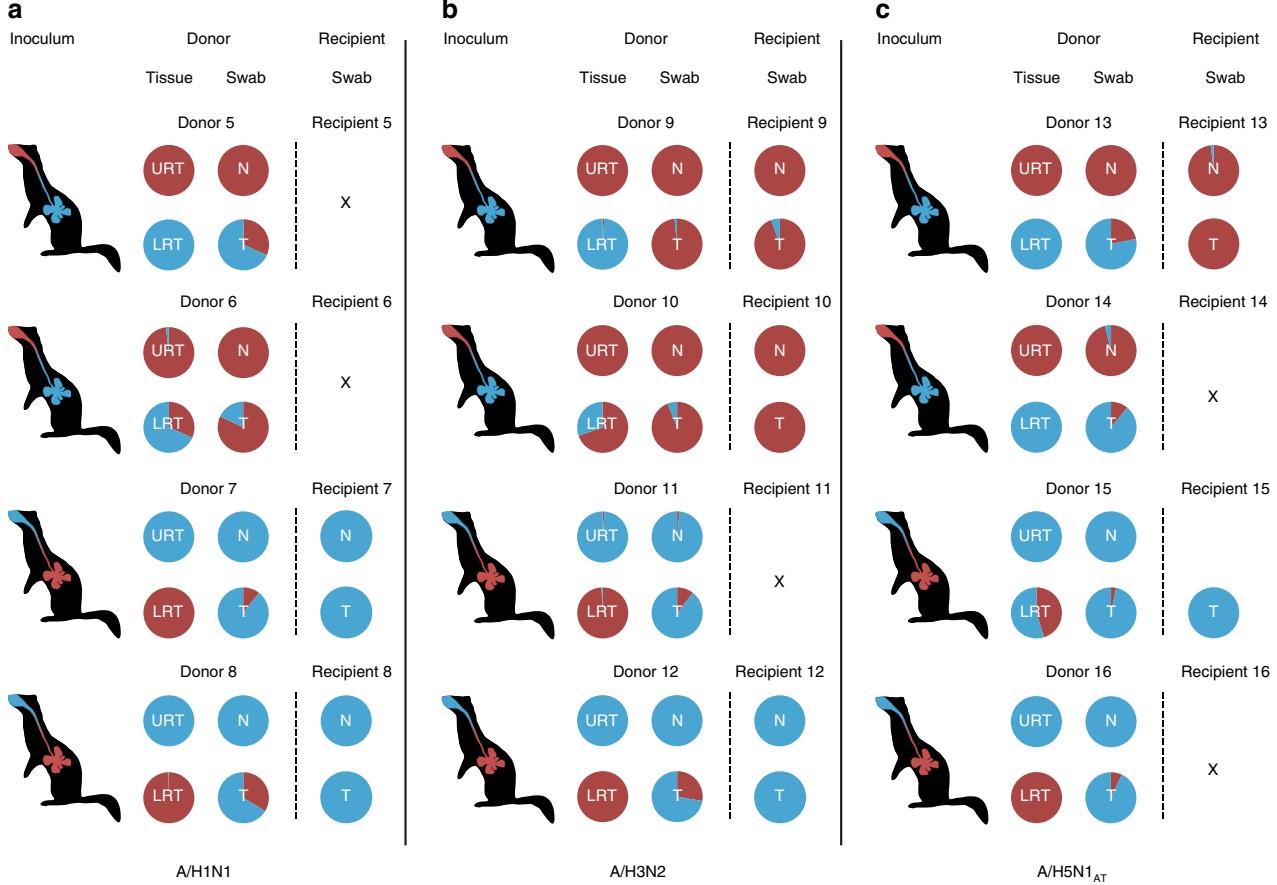

**Fig. 2 A/H1N1, A/H3N2 and A/H5N1_AT viruses were transmitted from the upper respiratory tract of ferrets.** Ferrets were inoculated with $10^5$ $TCID_{50}$ of untagged (shown in red) and tagged (shown in blue) virus pairs of A/H1N1 virus (**a** Donors 5–8), A/H3N2 virus (**b** Donors 9–12) or A/H5N1_AT virus (**c** Donors 13–16). Untagged and tagged viruses were inoculated either intranasally or intratracheally as indicated by the color coding in the schematic ferret representations. Untagged and tagged virus proportions at the day of virus transmission are represented by the pie charts for both donor and recipient ferrets. URT: upper respiratory tract, i.e., nasal turbinates, LRT: lower respiratory tract, i.e., combined data from two parts of the trachea and the lungs, N: nose swabs, T: throat swabs. X means that no transmission was observed. Transmission was defined by the detection of two consecutive swabs with a RT-qPCR threshold (CT-value) of 35. Source data are provided as a Source Data file.

nasal turbinates tissue (containing both nasal respiratory and olfactory epithelia), were determined. Data from our previous study on A/H1N1 and A/H3N2 viruses were included in the analysis[34]. At 2 dpi, airborne transmissible A/H1N1, A/H3N2 and A/H5N1_AT viruses had infected on average 55%, 43% and 37% of the nasal respiratory epithelium, whereas the A/H5N1 virus was detected in only 2% of the epithelium (Fig. 3a, b). At 1 dpi, 90% and 30% of the respiratory epithelium was infected by A/H1N1 and A/H3N2 viruses respectively (Fig. 3b), highlighting differences in infection kinetics between the two viruses.

At 2 dpi, the nasal olfactory epithelium was less infected by the airborne transmissible A/H1N1, A/H3N2 and A/H5N1_AT viruses than the nasal respiratory epithelium (40%, 0% and 7% respectively). In contrast, more cells of the nasal olfactory epithelium were infected by the A/H5N1 virus than of the respiratory epithelium (18% and 2% respectively). Given these observations, it appears unlikely that tropism in the nasal olfactory epithelium plays a role in airborne transmission of IAVs. Infectious virus titers in the nasal turbinates did not reflect these differences, partially because the homogenized nasal turbinate sample contained both nasal respiratory and olfactory epithelia (Fig. 3c). Mean virus titers in nasal turbinates of ferrets inoculated with A/H3N2 were lower than those in ferrets inoculated with A/H1N1, A/H5N1_AT and A/H5N1. Moreover,

despite infecting a lower percentage of cells, the A/H5N1 virus replicated to high virus titers in the nasal turbinates. Together, these data suggest that infection of the nasal respiratory epithelium, rather than replication to high titers in the nasal turbinates, is a driver of airborne transmission. The availability of genetically close viruses with distinct infection phenotypes in the nose of ferrets (A/H5N1_AT and A/H5N1) provided an excellent opportunity to investigate the substitutions that were responsible for the nasal respiratory epithelium tropism. To this end, subsets of mutations that are present in A/H5N1_AT were introduced into the A/H5N1 background. Recombinant viruses carrying either the substitutions in the polymerase and the nucleoprotein genes (A/H5N1_POL-mut, containing PB2-E627K, PB1-H99Y, PB1-I368V, NP-R99K, NP-S345N) or the substitutions in the HA gene (A/H5N1_HA-mut containing HA-H103Y, HA-T156A, HA-Q222L, HA-G224S (A/H5 numbering throughout the manuscript[35])) were produced using reverse genetics and used to inoculate three ferrets intranasally. Two days after inoculation, the animals were euthanized. The percentage of epithelium that was antigen-positive in the nasal respiratory and olfactory epithelia was scored and infectious virus titers in the homogenized nasal turbinates tissue were determined. Eighty-eight percent of the nasal respiratory epithelial cells of ferrets inoculated with A/H5N1_HA-mut were infected, against 13% for ferrets inoculated

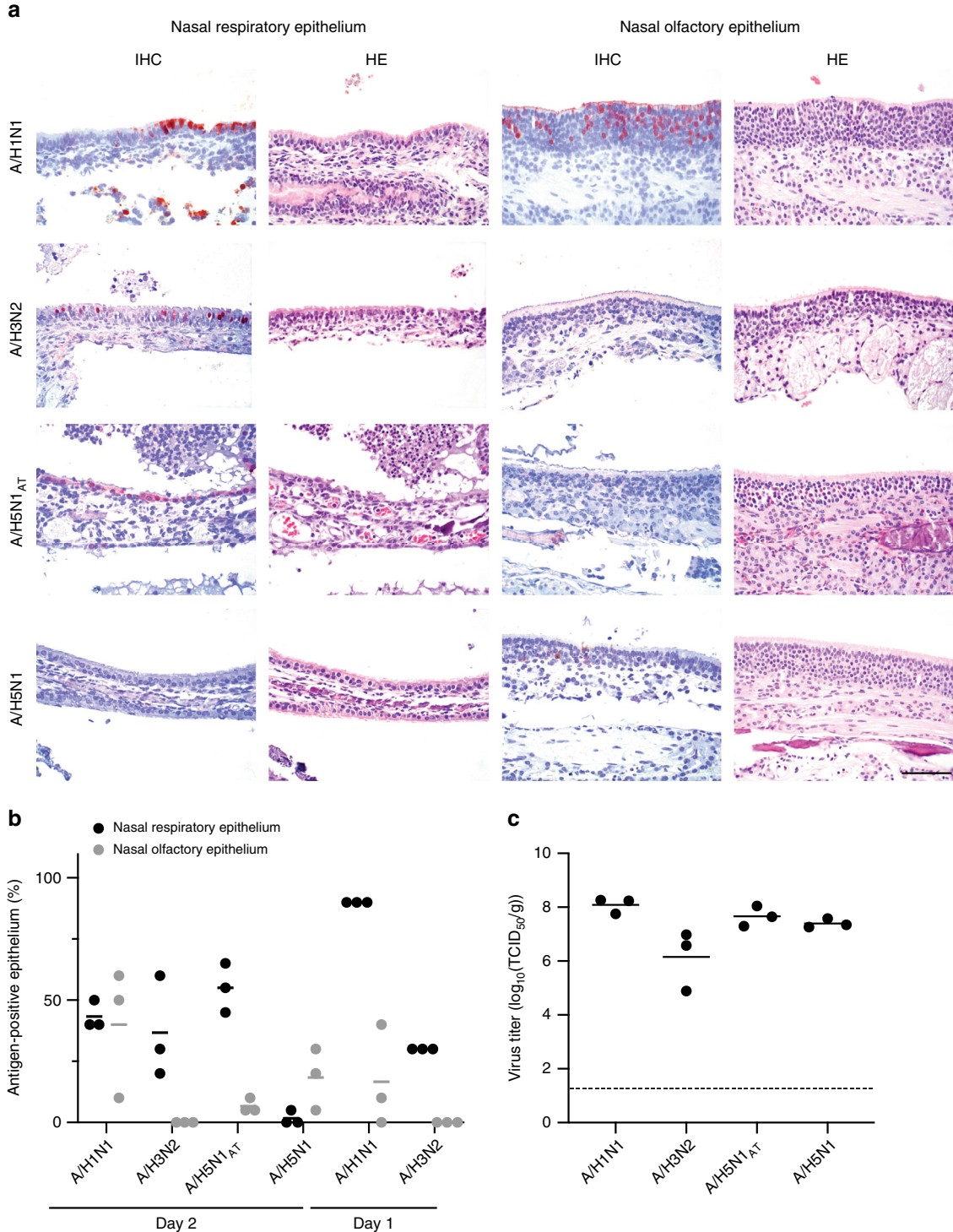

**Fig. 3 Airborne transmissible influenza A viruses infected the nasal respiratory epithelium of ferrets. a** Representative pictures of ferret nasal respiratory and nasal olfactory epithelia 2 days after intranasal inoculation with A/H1N1, A/H3N2, A/H5N1$_{AT}$ or A/H5N1 viruses. Influenza A virus nucleoprotein expression was determined by immunohistochemistry (IHC) and is shown as a red stain. HE: hematoxylin-eosin stain. Scale bar 50 μm. **b** Percentage of epithelium that was nucleoprotein antigen-positive, as determined by IHC, were blindly assessed in the nasal respiratory epithelium (black) and nasal olfactory epithelium (light gray) of three ferrets inoculated with the respective viruses. Individual percentages are shown. Means are depicted by the horizontal lines. **c** Individual virus titers in the homogenized nasal turbinates (containing both nasal respiratory and olfactory epithelia) were determined by end-point titration in MDCK. Means are depicted by the horizontal lines. The limit of detection of the virus titration is shown by the dotted line. Source data are provided as a Source Data file.

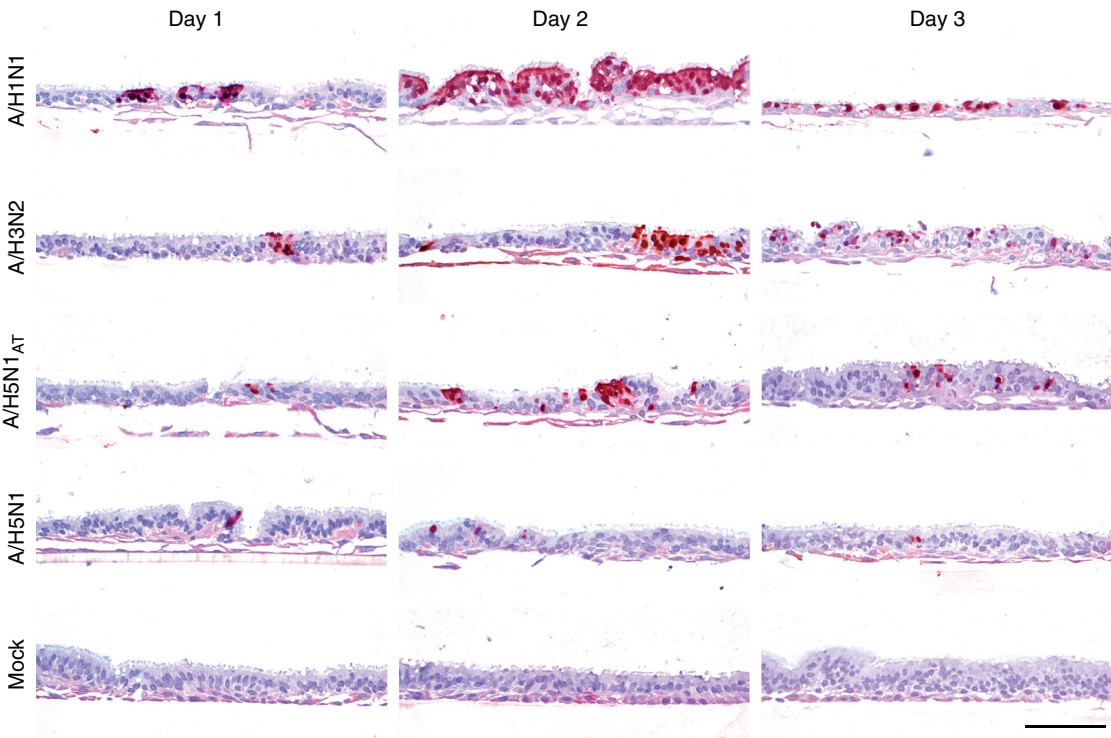

**Fig. 4 Airborne transmissible influenza A viruses infected the human respiratory epithelium.** Representative pictures of primary cultures of human nasal respiratory epithelium (Mucilair™) inoculated with A/H1N1, A/H3N2, A/H5N1ₐₜ, A/H5N1 viruses or PBS (Mock). Influenza A virus nucleoprotein expression was determined by immunohistochemistry (IHC) and is shown as a red stain. Scale bar 100 μm.

with A/H5N1$_{POL-mut}$ (Supplementary Fig. 5). Using combinations of the four HA substitutions, we subsequently found that receptor-binding substitutions Q222L/G224S alone were not sufficient to confer a similar nasal respiratory epithelium tropism as that of A/H5N1$_{AT}$. However, when Q222L and G224S were combined with either H103Y or T156A, the percentage of nasal respiratory epithelium that was infected was similar or higher to that of A/H5N1$_{AT}$ (Supplementary Fig. 5, Panel a and b). Comparable to our previous observations, all viruses replicated to similar titers in the nasal turbinates, independent of the percentage of infected nasal respiratory and olfactory epithelium.

**Airborne IAVs infect human nasal respiratory epithelial cells.** To allow extrapolation of results from ferrets to humans, it was investigated whether airborne transmissible IAVs also preferentially infect human nasal respiratory epithelial cells. Primary human nasal epithelial cells (Mucilair™) were purchased from Epithelix Sárl. These cells were isolated from 14 healthy donors who underwent polypectomy, pooled, and fully differentiated at the air-liquid interface on a transwell membrane for 45 days. They contain the cell types present in the nasal respiratory epithelium: ciliated cells, mucus-producing goblet cells and basal cells. However, the origin of the respiratory epithelium, i.e., nasal conchae or nasopharynx, is unknown. First, in order to understand whether these primary human nasal epithelial cells can be used as a model for human nasal epithelial tissue, the binding pattern of human (A/H1N1 and A/H3N2), avian (A/H5N1) and modified avian (A/H5N1$_{Q222L/G224S}$) influenza viruses to these cells was determined using virus histochemistry (Supplementary Fig. 6). Ferret nasal turbinates and duck colon tissues were used as controls for the binding of human and avian influenza viruses respectively. A/H1N1, A/H3N2, and A/H5N1$_{Q222L/G224S}$ viruses abundantly attached to the apical side of ciliated epithelial cells, as

previously described[21,27]. The A/H5N1 virus occasionally attached to ciliated epithelial cells, as observed in the human nasopharynx[27], suggesting that at least a proportion of the cells might be derived from human nasopharynx rather than human nasal conchae.

A/H1N1, A/H3N2, A/H5N1, and A/H5N1$_{AT}$ viruses were then used to inoculate the primary human nasal epithelial cells in duplicate. Two transwell membranes per virus were fixed in formalin 1, 2 or 3 days after inoculation and infected cells were stained by immunohistochemistry detecting the nucleoprotein (Fig. 4). The human A/H1N1 virus abundantly infected the ciliated nasal epithelial cells and, three days after inoculation, all ciliated epithelium was damaged due to the infection. The A/H3N2 virus infected the ciliated epithelial cells to a lesser extent than the A/H1N1 virus, as observed in the ferret nasal turbinates. However, by 3 dpi, part of the ciliated epithelium exhibited shortened or destroyed cilia, as the result of infection. In contrast, the A/H5N1 virus only infected ciliated epithelial cells occasionally, and the infection did not progress during the course of the experiment. The infection phenotype of the A/H5N1$_{AT}$ virus was intermediate to that of A/H3N2 and A/H5N1 viruses. These results showed that, as observed in ferrets, human airborne transmissible IAVs also abundantly infected in primary human nasal respiratory epithelial cells, in contrast to the non-airborne transmissible influenza A/H5N1 virus.

**Discussion**
IAVs can be transmitted via non-mutually exclusive routes of transmission: direct contact, indirect contact, respiratory droplets, or aerosols. However, the relative contribution of each route to efficient IAV transmission remains unknown and under debate. Respiratory droplet transmission is mediated by expelled particles that have a propensity to settle quickly because of their size and is

therefore reliant on close proximity between infected (donor) and susceptible (recipient) individuals, usually within 1 m of the site of expulsion[36]. Aerosol transmission is mediated by expelled particles that are smaller in size than respiratory droplets and can remain suspended in the air for prolonged periods of time, allowing infection of susceptible individuals at a greater distance from the site of expulsion. A generally accepted cut-off size to discriminate between respiratory droplets and aerosols is 5 μm diameter[37]. The current paradigm adopted by the Word Health Organization (WHO) is that influenza viruses are transmitted via respiratory droplets, when aerosol-generating procedures are excluded[37]. Therefore, current guidelines to prevent influenza virus transmission in health care settings are only based on preventing respiratory droplet transmission (summarized in ref. [38]). However, the recent body of work on the detection of influenza virus genomes and infectious particles in aerosols suggests that IAVs are also transmitted by aerosols[4–15]. This discordance between guidelines and experimental data highlights the urgent need to improve our fundamental understanding of influenza virus transmission. One current gap in this understanding is the identification of the anatomical site of the respiratory tract from which influenza virus-laden particles are generated and expelled for onwards transmission. It has been shown that preference of the virus to bind to α2,6-SA, associated with viral replication in the ciliated cells in the URT (nasal turbinates, pharynx, larynx) and part of the LRT (trachea, bronchus), is an important determinant for airborne transmissibility of IAVs[23–25]. However, it still remained unknown whether the source of exhaled viruses from the donor is the epithelium of tissues in the URT, the LRT or both, and whether α2,6-SA binding preference is necessary for the virus to be exhaled from the donor or for the virus to initiate replication in the recipient.

Here, it was shown for the first time that human A/H1N1 and A/H3N2 viruses and mammal-adapted avian A/H5N1 virus are transmitted via the air from the URT, more specifically from the nasal respiratory epithelium, and not from the trachea, bronchus or the lungs of inoculated ferrets. Transmission was delayed and less robust than observed in previous experiments, in which the donor ferret is only inoculated intranasally and stay in contact with the recipient ferret for 14 days (rather than maximum 5 days)[18,32]. The reason for this might be the fact that the donor ferrets were suffering from lower respiratory tract disease, possibly leading to impaired breathing, or that the donor ferrets were removed from the experiment in some cases too early.

The results of this study imply that replication in the URT of ferrets, and more specifically the nasal respiratory epithelium, is important for the generation and expulsion of influenza virus-laden particles from donor ferrets. As particles expelled from the URT are thought to be bigger than those expelled from the LRT, the results of this study could imply that transmission of IAVs between ferret is mediated by respiratory droplets rather than aerosols, which is in accordance with the studies by Zhou et al.[39] and Gustin et al.[40]. Zhou et al.[39] showed that naive or influenza-inoculated ferrets exhaled a greater number of fine particles (<1.5 μm) than large particles but that viral RNA was predominantly present in particles > 4 μm. Consistent with this observation, transmission between ferrets was abolished when particles > 1.5 μm were captured by a size impactor. Gustin et al.[40] also showed that viable virus detection in inoculated ferret breath was five times higher in particles > 4.7 μm than in particles <4.7 μm. Isolation of infectious virus from aerosols expelled by infected individuals[4,7,11,14] gave rise to the hypothesis that influenza virus-laden particles are more likely to originate from the deepest parts of the lungs, where small aerosol particles are hypothesized to be generated by the reopening of collapsed small airways during the previous inhalation[41–45], than from the upper

or oropharyngeal airways[4,6,7]. This hypothesis is in contradiction with the results of this study. However, evaporation of respiratory droplets generated in the upper respiratory tract could lead to transmission via aerosols. Collection of viable influenza virus from the air and accurate determination of particle size, because of the rapid evaporation of droplets and aerosols, remain difficult. Improvement of respiratory droplet and aerosol collection techniques that preserve the size and infectivity of the virus particles is greatly needed. Moreover, once influenza viruses are expelled from the donor, they must remain stable in aerosols/respiratory droplets to be able to initiate a new infection in the recipient. Chemo-physical properties of both the air and the particles, including temperature, ultraviolet radiation, humidity and air movement influence the virus stability and infectivity[46]. In addition, the rate of evaporation of aerosols is higher than that of droplets, which might impact virus survival. After the airborne phase, the size of aerosols and droplets will determine the region of deposition: whereas droplets tend to be deposited in the URT, aerosols can be inhaled and deposited deep in the LRT[47]. Moreover, deposition needs to take place in a part of the respiratory tract where appropriate receptors are expressed, which is the case for the URT where α−2,6-SA are prevalent[28,29]. The site of infection initiation in the recipient upon transmission via the air remains unknown. In ferret transmission experiments[18,21,32,48], throat swabs of recipient ferrets usually become virus-positive before nose swabs, suggesting that virus deposition and infection is initiated in the oropharyngeal cavity. This would be consistent with transmission being mediated by respiratory droplets that would be generated in the URT of donor ferrets and deposited in the URT of recipient ferrets. The soft palate, forming the floor of the nasopharynx in the oropharyngeal cavity, has been recently identified as a potential site for the selection of airborne transmissible viruses that bind to long-chain α−2,6-SA and it was suggested that it could be the initial site of infection upon airborne transmission[49].

Despite the observation that airborne transmissible IAVs are transmitted from the URT of inoculated ferrets, the exact site of particle generation within the URT remains unknown. In the first set of experiments, only the nasal turbinates were collected, and it cannot be excluded that airborne transmissible IAVs were transmitted from the throat (pharynx, larynx). However, even when the virus inoculated in the LRT was detected in throat swabs of donor ferrets, it was never transmitted as the dominant variant to recipient ferrets, suggesting that influenza virus-laden particles were not expelled from the oropharyngeal cavity of donor ferrets. All influenza viruses tested in this study replicated to high titers in the nasal turbinates of inoculated ferrets, however, clear differences in tropism within the nasal turbinates tissue were observed. A/H1N1, A/H3N2, and A/H5N1$_{AT}$ viruses abundantly infected the nasal respiratory epithelium of ferrets, contrary to avian A/H5N1 virus, supporting the hypothesis that airborne transmissible viruses are expelled from the nasal respiratory epithelium. Interestingly, although A/H5N1 virus only infected cells in the nasal olfactory epithelium, virus infectious titers in the homogenized nasal turbinate tissue were in the same range as those of A/H1N1 and A/H5N1$_{AT}$ viruses. In a recent study, infection of primary differentiated ferret nasal epithelial cells by avian A/H5N1 virus resulted in higher virus titers and more cell damage as compared to human A/H1N1 virus[50]. However, these differentiated primary ferret nasal epithelial cell cultures contained cells derived from both the respiratory and olfactory epithelia, potentially explaining why A/H5N1 virus replicated to higher titers than A/H1N1. In our study, the nasal respiratory epithelium tropism of A/H5N1$_{AT}$ virus was mediated by substitutions in the HA promoting binding to α−2,6-SA and/or stability. This change in tropism of the A/H5N1$_{AT}$ virus was

less obvious when assessed in primary human nasal respiratory epithelial cells, which might reflect the fact that A/H5N1$_{AT}$ virus was primarily adapted to transmit between ferrets. However, differences in infectivity in the human nasal respiratory epithelium between human and avian viruses were very clear, supporting the hypothesis that the nasal cavity could also be the preferred site for generation and expulsion of airborne transmissible influenza virus-laden particles in humans. Measuring the generation of influenza virus-laden particles upon breathing via the nose or the mouth separately would help in validating or inferring this hypothesis.

Here we propose a transmission model in which influenza virus-laden respiratory droplets are expelled from the nasal respiratory epithelium of the donor and deposited in the oropharyngeal cavity of the recipient. Virus replication is then initiated in the oropharyngeal cavity of the recipient after which the virus spreads to the nasal respiratory epithelium, from where it can be expelled for onwards transmission via the air. Should this model be correct, simple measures that target the URT to block transmission of IAVs could be implemented in health care settings. Continued efforts are necessary to fully understand the tropism of IAVs in humans in relation to airborne transmission, which will help to improve prevention measures in health care settings.

## Methods

**Cells**. Madin-Darby canine Kidney (MDCK) cells (ATCC) were cultured in Eagle's minimal essential medium (EMEM, Lonza Benelux BV, Breda, the Netherlands) supplemented with 10% foetal bovine serum (FBS) (Greiner), 100 U ml$^{-1}$ penicillin (PEN, Lonza), 100 U ml$^{-1}$ streptomycin (STR, Lonza), 2 mM L-glutamine (L-glu, Lonza), 1.5 mg ml$^{-1}$ sodium bicarbonate (NaHCO$_3$, Lonza), 10 mM Hepes (Lonza) and 1X non-essential amino acids (NEAA, Lonza). 293T cells (ATCC) were cultured in Dulbecco modified Eagle's medium (DMEM, Lonza) supplemented with 10% FBS, 100 U ml$^{-1}$ PEN, 100 U ml$^{-1}$ SRT, 2mM L-glu, 1 mM sodium pyruvate (Gibco) and 1X NEAA. Human airway epithelia reconstituted in vitro (MuciAir$^{TM}$, EP02MP) were purchased from Epithelix Sàrl (Switzerland).

**Viruses**. Untagged and tagged variant (var) recombinant A/Netherlands/602/2009 (A/H1N1) viruses were were initially described in ref. [30]. The tagged A/H1N1$_{var}$ virus carries a single synonymous nucleotide substitution per segment relative to the untagged A/H1N1 virus, as follows (nucleotide numbering is from the 5′ end of the cRNA): PB2 C273T, PB1 T288C, PA C360T, HA C305T, NP A351G, NA G336A, M G295A, NS C341T. These substitutions were introduced into the reverse genetics plasmids using QuikChange (Agilent) site directed mutagenesis, according to the manufacturer's instructions (for a complete list of the primers used in this study, see Supplementary Table 2). An 8 plasmid rescue system based on a modified version of pHW2000[51] and co-culture of 293T and MDCK cells were used. Plaque isolates derived from rescue supernatants were amplified in MDCK cells to generate virus stocks and stock titers were determined by endpoint titration in MDCK cells.

Untagged and tagged variant (var) recombinant A/Panama/2007/99 (A/H3N2) viruses were initially described in ref. [33]. The tagged A/H3N2$_{var}$ virus contains the following synonymous nucleotide substitutions relative to the untagged A/H3N2 virus: PB2 C354T, C360T; PB1 A540G; PA A342G, G333A; HA T308C, C311A, C314T, A464T, C467G, T470A; NP C537T, T538A, C539G; NA C418G, T421A, A424C; M G586A; NS C329T, C335T, A341G. These mutations were introduced into the pPOL1 reverse genetics plasmids using QuikChange (Agilent) site directed mutagenesis, according to the manufacturer's instructions. For A/H3N2 viruses, a 12 plasmid rescue system based on pPOL1 and pCAGGS vectors and co-culture of 293T and MDCK cells were used. Plaque isolates derived from rescue supernatants were amplified in 11-day-old embryonated chicken eggs incubated at 33ºC to generate virus stocks and stock titers were determined by endpoint titrations in MDCK cells.

Recombinant airborne transmissible A/H5N1 A/Indonesia/05/2005 virus (A/H5N1$_{AT}$) was initially described in ref. [21]. The A/H5N1$_{AT}$ virus carries 9 amino acid substitutions as compared to the A/H5N1 virus (A/H5N1): PB2-E627K, PB1-H99Y, PB1-I368V, NP-R99K, NP-S345N, HA-H103Y, HA-T156A, HA-Q222L, and HA-G224S. The var A/H5N1$_{AT}$ virus carries one silent nucleotide substitution in the PB2 gene (A339G), which was introduced into the reverse genetics plasmids using QuikChange (Agilent) site directed mutagenesis, according to the manufacturer's instructions. Recombinant A/H5N1 viruses carrying subsets of airborne substitutions were initially described in ref. [21]. Airborne substitutions were introduced into the reverse genetics plasmids using QuikChange (Agilent) site directed mutagenesis, according to the manufacturer's instructions. A/H5N1

viruses used for virus histochemistry were recombinant viruses with seven gene segments of A/Puerto-Rico/8/1934 and the wild-type or mutant HA segment (A/H5N1$_{HA-Q222L/G224S}$) of A/Indonesia/5/2005 from which the multibasic cleavage site was removed. Recombinant A/H5 influenza viruses were rescued in 293T cells by using reverse genetics using a modified version of pHW2000[51]. Cells were plated the day before transfection in gelatinized 100 mm diameter culture dishes to obtain 50% confluent monolayers. 293T cells were then transfected using calcium phosphate with 40 µg of total DNA. After overnight transfection, the transfection medium was replaced with fresh medium supplemented with 2% FCS for virus production. Cells were incubated for 72 h, after which supernatants were harvested. Virus-containing supernatants were cleared by centrifugation for 10 min at 300 × g and used to infect MDCK to generate virus stocks and stock titers were determined by endpoint titrations in MDCK cells. The A/H3N2 A/Netherlands/213/2003 and A/H1N1 A/Netherlands/602/2009 viruses used for virus histochemistry were human isolates propagated in MDCK cells.

**Biosafety**. Experiments with A/H1N1 and A/H3N2 viruses were performed under biosafety level 3 conditions and experiments with A/H5N1 were performed under biosafety level 3+ conditions. Experiments with A/H5N1$_{AT}$ were conducted in adherence with the conditions of the U.S. Government Gain-of-Function Deliberative Process and Research Funding Pause of Selected Gain-of-Function Research involving Influenza, MERS and SARS viruses[52].

**Ferret experiments**. All relevant ethical regulations for animal testing have been complied with. Animals were housed and experiments were performed in strict compliance with European guidelines (EU Directive on Animal Testing 86/609/EEC) and Dutch legislation (Experiments on Animals Act, 1997). Influenza virus and Aleutian Disease Virus seronegative 6-month-old female ferrets (*Mustela putorius furo*), weighing 700–1000 g, were obtained from commercial breeders (Euroferret (Denmark) and TripleF (USA)). All animal experiments received ethical approval from the independent animal experimentation ethical review committee 'stichting DEC consult' (Erasmus MC permit number 122-11-30 and 122-14-13). The DEC considers the application and pays careful attention to the effects of the intervention on the animal, its discomfort and weighs this against the social and scientific benefit to humans or animals. The researcher is required to keep the effects of the intervention to a minimum, based on the three Rs (Refinement, Replacement, Reduction). Animal welfare was monitored on a daily basis. Virus inoculation of ferrets was performed under anesthesia with a mixture of ketamine/medetomidine (10 and 0.05 mg kg$^{-1}$ respectively) antagonized by atipamezole (0.25 mg kg$^{-1}$). All animal handling (swabbing and weighing) was performed under light anesthesia using ketamine to minimize animal suffering.

(i) Ferret transmission experiments: Donor ferrets were inoculated intratracheally with 10$^5$ TCID$_{50}$ of virus diluted in a 3 ml volume of phosphate-buffered saline (PBS) and subsequently intranasally with 10$^5$ TCID$_{50}$ of virus diluted in a 40 µl volume of PBS (20 µl instilled in each nostril). Half of the donor ferrets were inoculated with untagged viruses intranasally and tagged viruses intratracheally and the other half with the opposite placement of virus in order to correct for potential small differences in transmissibility (see Supplementary Table 1 for a summary). Throat and nose swabs were collected from donor ferrets every day until 7 days post inoculation (dpi) (donor ferret 1–4) or until the day transmission was observed or the latest at 5 dpi (donor ferrets 5–16). Recipient ferrets were placed four hours after inoculation (hpi) of donor ferrets in an opposite cage separated by two steel grids, 10 cm apart, to avoid contact transmission. Throat and nose swabs were collected from recipient ferrets 12 h respectively until 9 days post exposure (dpe). Swabs were stored at −80 °C in transport medium (Hanks' balanced salt solution containing 0.5% lactalbumin (Sigma-Aldrich), 10% glycerol (Sigma-Aldrich), 200 U ml$^{-1}$ PEN, 200 mg ml$^{-1}$ STR, 100 U ml$^{-1}$ polymyxin B sulfate (Sigma-Aldrich), and 250 mg ml$^{-1}$ gentamicin (Gibco)) for end-point titration in MDCK and next-generation sequencing as decribed below. Shedding from recipient ferrets 5–16 was monitored every day by performing real-time RT-qPCR detecting the matrix gene on nose and throat swabs right after collection as described below. The rest of the swabs were stored in transport media at −80 °C for endpoint titration in MDCK cells and next-generation sequencing. The corresponding donor ferrets were euthanized by heart puncture under anesthesia when transmission to the recipient ferret was observed (two consecutive positive swabs with a threshold value in RT-qPCR (CT value) < 35) or at 5 dpi the latest. Samples from the respiratory tract (nasal turbinates, upper part of the trachea, lower part of the trachea, left lung lobes and right lung lobes) were collected, homogenized in transport medium using a FastPrep system (MP Biomedicals) with 2 one-quarter-inch ceramic sphere balls, centrifuged at 1500 × g for 10 min, aliquoted, and stored at −80 °C for endpoint titration in MDCK cells and next-generation sequencing. Additionally, throat and nose swabs were collected from donor ferrets 5–16 every day until euthanasia and stored in transport media at −80 °C for endpoint titration in MDCK cells and next-generation sequencing. Clonality of the virus inoculum was confirmed by next-generation sequencing. Virus inocula were back titrated to ensure that the right doses were used to inoculate donor ferrets.

(ii) Ferret infection experiments: Three ferrets per group were inoculated intranasally with a total dose of 10$^6$ TCID$_{50}$ of virus by instillation of 250 µl of virus suspension dropwise in each nostril. The A/H5N1 viruses tested carried subsets of

airborne substitutions: A/H5N1$_{POL-mut}$ (PB2-E627K, PB1-H99Y, PB1-I368V, NP-R99K, NP-S345N), A/H5N1$_{HA-mut}$ (HA-H103Y, HA-T156A, HA-Q222L and HA-G224S), A/H5N1$_{HA-Q222L/G224S}$, A/H5N1$_{HA-H103Y/Q222L/G224S}$, A/H5N1$_{HA-T156A/Q222L/G224S}$. Two days after inoculation, ferrets were euthanized by cardiac puncture and nasal turbinates were harvested. The left nasal turbinates were fixed in 10% neutral-buffered formalin, embedded in paraffin and sectioned at 3μm for immunohistochemical analysis. The right nasal turbinates were homogenized in transport medium using a FastPrep system (MP Biomedicals) with 2 one-quarter-inch ceramic sphere balls, centrifuged at $1500 \times g$ for 10 min, aliquoted, and stored at −80 °C for endpoint titration in MDCK cells.

**Immunohistochemistry**. Sequential slides of nasal turbinates were deparaffinised in xylene and hydrated using graded alcohols. They were stained with hematoxylin and eosin (HE staining) or for the detection of the IAV nucleoprotein as described here. Antigen retrieval was performed using a 0,1% solution of the protease from *Streptomycus griseus* (Sigma-Aldrich) in PBS for 10 min at 37 °C. After a wash in PBS, endogenous peroxidases were blocked by using a solution of 3% $H_2O_2$ in PBS for 10 min at room temperature. After one wash in PBS and one wash in PBS-0.05% Tween, slides were incubated with a monoclonal antibody against IAV nucleoprotein (mouse IgG2a anti-influenza A nucleoprotein, H16-L10-4R5 (ATCC® HB-65™) diluted 1/400 in PBS-0,1% bovine serum albumin (BSA) or with an isotype control (mouse IgG2a, MAB003, R&D Systems) diluted 1/200 in PBS-0.1% BSA for an hour at room temperature. After two washes in PBS-0.05% Tween, slides were incubated with a secondary antibody goat anti-mouse IgG2a coupled to horseradish peroxidase (HRP) (Biorad, Star133P) diluted 1/100 in PBS-0.1% BSA for an hour at room temperature. After two washes with PBS, HRP was revealed using 3-Amino-9-Ethylcarbazole (AEC, Sigma-Aldrich) in N,N-dimethylformamide (Honeywell Fluka) diluted in a final concentration of 0.0475 M of sodium acetate (NaAc, pH = 5) with 0.05% of $H_2O_2$ for 10 min at room temperature, resulting in a bright red precipitate. A counterstain was performed with hematoxylin and the slides were embedded using Kaiser's glycerol gelatin (Merck). In each staining procedure, a lung section from a cat infected experimentally with an A/H5N1 virus was used as a positive control. Immunohistochemical analyses were performed blindly by a veterinary pathologist. One slide with all the nasal tissue (including both respiratory and olfactory epithelium) was analysed per ferret. For the scoring, all the fields available on each slide were analysed and a percentage score was estimated for each slide. Pictures were taken using an Olympus BX41 microscope, an Olympus DP27 camera and acquisition Olympus CellSens entry software. The white balance of the pictures was adjusted using Adobe Photoshop.

**Next-generation sequencing**. Viral RNA was extracted from respiratory swab samples collected from donor or recipient ferrets and from organs of donor ferrets, using the High Pure RNA Isolation kit (Roche) according to the manufacturer's instructions. RNA was subjected to reverse-transcription using Superscript III (Invitrogen) and the following primer: AGCRAAAGCAGG. Amplicons from the PB2 genes were generated by PCR from the cDNA using the following primers: A/H1N1 (CGCACTCAGAATGAAGTGGA (F), GCCGAAGGTACCATGTTTCA (R), amplicon size of 265 nucleotides), A/H3N2 (CATAGTAGTGCAGAAAT GGTTCCGGAGAGA (F), CATAGTAGTGTTCGGCGTATCTTGACTTGA (R), amplicon size of 239 nucleotides) and A/H5N1 (CATAGTAGTGTGGAGCAAG ACAAATGATGC (F), CATAGTAGTGCTCCCACTTCATTTGGGAAA (R), amplicon size 288 nucleotides). These fragments were sequenced using the Roche 454 GS Junior sequencing platform. The fragment library was created for each sample according to the manufacturer's protocol without DNA fragmentation (GS FLX Titanium Rapid Library Preparation, Roche). The emulsion PCR (Amplification Method Lib-L) and GS Junior sequencing run were performed according to instructions of the manufacturer (Roche). Sequence reads from the GS Junior sequencing data were sorted by barcode and aligned to reference sequences using CLC Genomics software 6.0.2. The sequence reads were trimmed at 30 nucleotides from the 3′ and 5′ ends to remove all primer sequences. For quality control, sequence reads were trimmed for Phred scores of less than 20. The threshold for the detection of single nucleotide polymorphisms was manually set at 1% of the total number of reads per sample. Results were then expressed as percentage of the sum of the reads corresponding to the two possible nucleotides (untagged and tagged).

**Virus titrations**. MDCK cells were inoculated with 10-fold serial dilutions of virus stocks, nose swabs, throat swabs, or homogenized tissue samples. The cells were washed with PBS 1 h after inoculation and cultured in infection medium, consisting of EMEM supplemented with 100 U ml$^{-1}$ PEN, 100 U ml$^{-1}$ STR, 2 mM L-Glu, 1.5 mg ml$^{-1}$ NaHCO$_3$, 10 mM HEPES, 1× NEAA, and 20 μg ml$^{-1}$ trypsin (N-tosyl-l-phenylalanine chloromethyl ketone [TPCK]-treated trypsin; Sigma-Aldrich). Three days after inoculation, supernatants of cell cultures were tested for agglutinating activity using turkey red blood cells (TRBCs) as an indicator of virus replication. Infectious virus titers were calculated from four replicates each of the homogenized tissue samples, nose swabs, and throat swabs and from ten replicates of the virus stocks by the method of Reed and Muench[53].

**Real-time RT-PCR targeting the matrix gene**. Viral RNA extraction was performed using the High the High Pure RNA Isolation kit (Roche) according to the manufacturer's instructions. Real-time RT-PCR was performed using the TaqMan™ Fast Virus 1-Step Master Mix (ThermoFisher Scientific) and the following forward primer, reverse primer and probe: AAGACCAATCCTGTCACCTCTGA, CAAA GCGTCTACGGCTGCAGTCA and 6-FAM TTTGTGTTCACGGCTCACCGTGCC-T AMRA. Amplification and detection was performed on an ABI7700 (Thermo-Fischer Scientific) using the following program: 5 min 50 °C, 20" 95 °C, [3" 95 °C, 31" 60 °C] × 45 cycles.

**Infection of primary human nasal respiratory epithelial cells**. Human airway epithelia reconstituted in vitro (MucilAir$^{TM}$) were purchased from Epithelix Sàrl (Switzerland). These human respiratory epithelia were reconstituted from nasal polyps obtained from patients undergoing surgical nasal polypectomy. At Epithelix Sàrl, mixtures of cells originating from 14 donors were seeded in Transwell-COL inserts and cultured at the air-liquid interface. After 45-days of culture, the epithelia became fully differentiated with a pseudo-stratified architecture with the three main types of cells: ciliated epithelial cells, mucus-producing goblets cells and basal cells. Cells were tested negative for mycoplasma, HIV-1, HIV-2, hepatitis B and C. Cells were received when fully differentiated, were cultured at 37 °C, 5% CO$_2$ and basal media was changed every two days. The total number of differentiated cells was estimated to be 400,000 cells per well. Before the inoculation, cells were incubated with PBS supplemented with Ca$^{2+}$ and Mg$^{2+}$ (100 mg/L of CaCl$_2$ and 100 mg/L of MgCl$_2$−6H$_2$0) for 45 min at 37 °C, 5%CO$_2$ and subsequently washed three times with PBS with Ca$^{2+}$ and Mg$^{2+}$. This treatment tightens the junctions and removes the mucus accumulated at the apical surface of the cells. Cells were then inoculated in duplicates with A/H1N1, A/H3N2 (A/Panama/2007/ 99), A/H5N1 or A/H5N1$_{AT}$ viruses at a multiplicity of infection (m.o.i) of 0.1 TCID$_{50}$/cell for three hours at 37 °C, 5%CO$_2$. Cells were then washed three times with PBS with Ca$^{2+}$ and Mg$^{2+}$ and then left at the air-liquid interface. At 1, 2 and 3 days after inoculation, cells were washed 5 times with PBS with Ca$^{2+}$ and Mg$^{2+}$ and fixed in 10% buffered formalin, embedded in paraffin and sectioned at 3μm for immunohistochemical analysis as described above for the ferret nasal respiratory and olfactory epithelia. Each membrane was sliced at three different positions to have a representation of the infection pattern on the overall membranes. Representative pictures were taken using an Olympus BX51 microscope, an Olympus ColorView IIIu camera and acquisition Olympus Cell$^A$ software. The white balance of the pictures was adjusted using Adobe Photoshop.

**Virus histochemistry**. The pattern of virus attachment to human nasal respiratory epithelium was determined by virus histochemistry[54]. Formalin-fixed, paraffin-embedded sections from three uninfected control wells were deparaffinised in xylene and hydrated using graded alcohols. Endogenous peroxidases were blocked with 3% H$_2$O$_2$ diluted in PBS for 10 min at room temperature. After two washes with PBS, a blocking step with a Tris-NaCl-blocking buffer (TNB buffer, 0.5% of blocking reagent (Perkin Elmer) in 0.1 M Tris HCl, 0.15 M NaCl, pH = 7.5) for 30 min at room temperature was performed. Hundred hemagglutination units of fluorescin isothiocyanate (FITC)-labeled influenza viruses (A/H1N1, A/H3N2 A/ Netherlands/213/2003, A/H5N1, A/H5N1$_{HA-Q222L/G224S}$) were incubated on the slides overnight at 4 °C in TNB buffer. After two washes with PBS-0.05% Tween, FITC was detected with a peroxidase-labelled rabbit–anti-FITC (DAKO, P5100) diluted 1/100 in TNB buffer for 1 h at room temperature. After two washes with PBS-0.05% Tween, the signal was amplified using a tyramide amplification system (Perkin-Elmer) according to the manufacturer's instructions. After two washes with PBS-0.05% Tween, slides were incubated with HRP coupled anti-streptavidin antibody (DAKO, D0397) diluted 1/300 in TNB buffer for 30 min at room temperature. After two washes with PBS, HRP was revealed using 3-Amino-9-Ethylcarbazole (AEC, Sigma-Aldrich) in N,N-dimethylformamide (Honeywell Fluka) diluted in a final concentration of 0.0475 M of sodium acetate (NaAc, pH = 5) with 0.05% of H$_2$O$_2$ for 10 min at room temperature, resulting in a bright red precipitate. A counterstain was performed with hematoxylin and the slides were embedded using Kaiser's glycerol gelatin (Merck). Ferret nasal turbinates and duck colon were included as controls for binding of human and avian viruses respectively. Pictures were taken using an Olympus BX51 microscope, an Olympus ColorView IIIu camera and acquisition Olympus Cell$^A$ software. The white balance of the pictures was adjusted using Adobe Photoshop.

**Reporting summary**. Further information on research design is available in the Nature Research Reporting Summary linked to this article.

## Data availability
Data underlying Figs. 1, 2, 3b, 3c and supplementary Figs. S2, S3, S4, S5b, S5c are provided as Source Data files. All other data are available from the corresponding author (S.H.) on reasonable request.

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

## Acknowledgements

We thank Peter van Run for technical assistance with histochemistry. We thank Debby van Riel and Thijs Kuiken for constructive discussions. This work was supported by NIH/NIAID contracts HHSN272201400008C and HHSN27220140004C. S.H. was funded in part by an NWO VIDI grant (contract number 91715372).

## Author contributions

M.R., J.M.A.v.d.B. and S.H. conceived, designed, analysed and performed the work. M.R. and S.H. wrote the manuscript. T.M.B., P.L. and D.M. helped with performing the work. R.A.M.F. and A.C.L. helped with the design of the work, interpretation of the data and manuscript revision. All authors read and approved the final manuscript.

## Competing interests

The authors declare no competing interests.
