## [Peer Review File · Nature Communications]

Reviewers' comments:

Reviewer #1 (Remarks to the Author):

In this manuscript "Influenza A viruses that are transmissible via the air are expelled from the upper respiratory tract of ferrets upon replication in the nasal respiratory epithelium", Herfst et al use an innovative approach to demonstrate that airborne transmission of influenza viruses in ferrets is via viruses replicating in the upper respiratory tract. This is independent of viral strain as H1, H3 and transmission-positive H5 viruses behave similarly. They used primary human nasal respiratory cells *ex vivo* to extrapolate their findings to humans. The studies are well-designed and the results support the author's conclusions. There is sufficient description of the methods to allow others to reproduce the work. These are novel and important findings for the influenza community as well as other groups interested in transmission of respiratory agents. They definitively prove what others have speculated. Beautiful and important study.

Reviewer #2 (Remarks to the Author):

Richard et al extend an experimental approach presented previously in PMID 29212934, where they inoculate ferrets with wild-type and genetically tagged variant viruses at discrete anatomical sites, to examine compartmentalized virus replication in the ferret nasal passages and lung. In this study, they employ these dual-virus infected ferrets in respiratory droplet transmission experiments, to examine if virus deposited in the nasal passages or lung is transmitted to naïve contact ferrets. Three pairs of wild-type/tagged virus pairs (H1N1, H3N2, H5N1-mut) are tested in this manner. Additionally, histopathology readouts of ferret tissues and human respiratory tract cells are included. The authors find that virus replication in the ferret nasal passages, and not the lung, represents the source of virus transmitting to contact animals, and identify a role for the ferret nasal respiratory (and not olfactory) epithelium in this event. The study is well-written, with results logically organized and presented. Furthermore, the questions under investigation in this study represent important ones for the field. However, there are areas of this manuscript that would benefit from additional clarification or contextualization. Furthermore, some of the conclusions and implications of this work as presented in the discussion are a bit outside the scope of data presented in the results.

Major comments:

1. Histopathology data (IHC and HE) presented in figures 3 and 4 (ferret nasal/olfactory epithelium and human respiratory epithelium) is difficult to fully interpret in the absence of infectious viral titers presented concurrently. Can the authors provide some measure of infectious viral load for the tissues collected and presented in Figure 3 (even if it is one combined nasal turbinates sample – this is important as ferrets in Figure 1 and 2 were inoculated with a 40ul volume but ferrets in Figure 3 were inoculated with a 500ul volume, as such authors cannot easily extrapolate viral replication kinetics between these experiments)? As the authors use this histopathology data to “highlight differences in infection kinetics” (lines 178-9), without infectious virus concurrently presented, this conclusion is not as strong as it could be. Similarly, for human respiratory epithelial cell data presented in Figure 4, influenza nucleoprotein staining alone captures an incomplete picture of virus replication. Authors should include infectivity data (even if it is just limited to supernatants collected on the days of viral staining as there were likely not sufficient numbers of cells to conduct a full replication curve), or temper their interpretation of results to focus on “infectivity” and not “replication”.

2. Throughout the study, the authors use “URT” and “LRT” to differentiate between virus replication in different anatomical sites in the ferrets, however, this can be misleading and fails to capture some of the novelty of this work. In example, on lines 61-62 and 282-3 the authors state that this study is the first to show that virus replication in the URT of inoculated ferrets is the site of replication of transmitting influenza viruses. However, numerous other studies employing pre-pandemic H1N1 and seasonal H3N2 viruses (which typically are restricted to the URT for replication and do not replicate in lower respiratory tract tissues, including the lung) have already demonstrated a role for the URT in virus transmission. What makes this study unique (in this reviewer’s opinion) is identification of virus replication in nasal turbinates samples (in the potential absence of other tissues such as the trachea), and identification of potential cell types within the milieu of the nasal turbinates, in this property. It would likely be more precise to rephrase (in example, line 145, though there are several instances of this) that virus replication in nasal turbinates, and not “URT”, is associated with transmission. Furthermore, it is confusing to use “LRT” to define lower respiratory tract as not just the lung but also the entirety of the trachea – is an upper trachea sample truly representative of the “lower respiratory tract”? The authors were smart to collect multiple sections of both the trachea and lung in their studies; it is very unclear why they are not capitalizing on this and using more precise language throughout this study.

3. The authors need to more clearly state how virus transmission is defined in this study. For H5N1-mut data shown in Figure 2 and supplemental Figure 3, the transmission data presented is not very convincing as presented. Recipient 15 does not appear to have any detectable infectious virus in nose or throat swabs (only nose swabs positive via PCR per supplemental data for this ferret, though Figure 3 indicates a T and not N, so this is unclear), and Recipient 13 has only one sample with positive infectious virus detection in the throat swab only; the nose is only positive via PCR. Specifically for Recipient 15, it seems very misleading to the reader to present data in Figure 3 highly suggesting virus detection in throat and nose swabs yet have this be via PCR only and not viral titers?

This is not standard practice in the field. In this reviewer's opinion, the H5N1 virus did not exhibit robust transmission in any pair, with only Recipient 13 possessing infectious virus in 1 of 2 collected samples, and the other 3 recipients not possessing infectious virus in nose or throat swabs collected at any time. Authors should state criteria for transmission definition in the text and temper their interpretation of transmission data from this component of the study accordingly.

4. The authors dedicate a substantial portion of the discussion towards the implications of this study in aerosol transmission and aerobiology, yet do not include any aerobiology data in this study. While inclusion of aerobiology-based text in the discussion is warranted, the absence of aerobiology data supporting this text makes the large focus in the discussion of this topic feel especially speculative, and again (in this reviewer's opinion) diminishes the true novelty of this work.

Minor comments:

1. Collection of discrete nose and throat swabs are not frequently collected samples in this field (compared with nasal washings). It would be of benefit for the reader to include additional text in the methods or results regarding the anatomical positioning of these samples (especially the throat sample, and the relative proximity of this specimen in relation to the soft palate), so the reader can more readily interpret this data. Similarly, do the nose swabs collected encapsulate virus replication from both respiratory and olfactory epithelia? The authors discuss ex vivo ferret cultures inclusive of both cell types in the discussion (lines 340-345) and it is unclear to the reader how to contextualize infectious virus from nose swabs in this argument without a precise understanding of how this sample is collected.

2. Lines 96-7, please provide the percentage of virus detected in throat swabs following intratracheal inoculation similar to how this is presented in lines 101-102 – it is difficult to see visually (specifically for donor 3) in Figure 1.

3. Specify in figure legends for Fig 1 and supplementary data that gray bars represent infectious virus.

4. Lines 116-7, "...evidence of active replication in the LRT was missing." Is this statement referring to data from previous studies (in which case, provide a reference) or data not shown (in which case, specify this)? This statement seems to contradict with line 136 that states that virus instilled following intratracheal inoculation was indeed replicating.

5. Please provide a supporting reference for the statement on lines 259-61 regarding distance of particles expelled.

6. Methods: please specify the order in which dual-virus ferret inoculations took place (did intranasal instillation occur before or after intratracheal instillation of virus)? Also specify the vendor of ferrets in this study.

Reviewers' comments:

Reviewer #1 (Remarks to the Author):

In this manuscript "Influenza A viruses that are transmissible via the air are expelled from the upper respiratory tract of ferrets upon replication in the nasal respiratory epithelium", Herfst et al use an innovative approach to demonstrate that airborne transmission of influenza viruses in ferrets is via viruses replicating in the upper respiratory tract. This is independent of viral strain as H1, H3 and transmission-positive H5 viruses behave similarly. They used primary human nasal respiratory cells ex vivo to extrapolate their findings to humans. The studies are well-designed and the results support the author's conclusions. There is sufficient description of the methods to allow others to reproduce the work. These are novel and important findings for the influenza community as well as other groups interested in transmission of respiratory agents. They definitively prove what others have speculated. Beautiful and important study.

We thank the reviewer for their very positive comments about our manuscript.

Reviewer #2 (Remarks to the Author):

Richard et al extend an experimental approach presented previously in PMID 29212934, where they inoculate ferrets with wild-type and genetically tagged variant viruses at discrete anatomical sites, to examine compartmentalized virus replication in the ferret nasal passages and lung. In this study, they employ these dual-virus infected ferrets in respiratory droplet transmission experiments, to examine if virus deposited in the nasal passages or lung is transmitted to naïve contact ferrets. Three pairs of wild-type/tagged virus pairs (H1N1, H3N2, H5N1-mut) are tested in this manner. Additionally, histopathology readouts of ferret tissues and human respiratory tract cells are included. The authors find that virus replication in the ferret nasal passages, and not the lung, represents the source of virus transmitting to contact animals, and identify a role for the ferret nasal respiratory (and not olfactory) epithelium in this event. The study is well-written, with results logically organized and presented. Furthermore, the questions under investigation in this study represent important ones for the field. However, there are areas of this manuscript that would benefit from additional clarification or contextualization. Furthermore, some of the conclusions and implications of this work as presented in the discussion are a bit outside the scope of data presented in the results.

Major comments:

1. Histopathology data (IHC and HE) presented in figures 3 and 4 (ferret nasal/olfactory epithelium and human respiratory epithelium) is difficult to fully interpret in the absence of infectious viral titers presented concurrently. Can the authors provide some measure of infectious viral load for the tissues collected and presented in Figure 3 (even if it is one combined nasal turbinates sample – this is important as ferrets in Figure 1 and 2 were inoculated with a 40ul volume but ferrets in Figure 3 were inoculated with a 500ul volume, as such authors cannot easily extrapolate viral replication kinetics between these experiments)? As the authors use this histopathology data to “highlight differences in infection kinetics” (lines 178-9), without infectious virus concurrently presented, this conclusion is not as strong as it could be. Similarly, for human respiratory epithelial cell data presented in Figure 4, influenza nucleoprotein staining alone captures an incomplete

picture of virus replication. Authors should include infectivity data (even if it is just limited to supernatants collected on the days of viral staining as there were likely not sufficient numbers of cells to conduct a full replication curve), or temper their interpretation of results to focus on “infectivity” and not “replication”.

We agree with the reviewer and therefore we added an additional panel to Figure 3 (panel c) providing information on infectious virus titers in the homogenized nasal turbinates tissue (combined sample containing both the respiratory and olfactory epithelia) of ferrets inoculated with A/H1N1, A/H3N2, A/H5N1 and A/H5N1_{AT}. For completeness, we also added an additional panel to Figure S5 (panel c) providing information about infectious virus titers in the homogenized nasal turbinates tissue of ferrets inoculated with the different A/H5N1 mutant viruses. Description of these data was added to the results section (lines 191-193, 218-225, 236-237, 243-245) and discussion (lines 369-376).

Unfortunately, we did not collect supernatants of human primary respiratory cells as these were cultured at the air-liquid interface and membranes were directly transferred to formalin for fixation and immunohistochemical analyses. However, we agree with the reviewer that our interpretation of the result should be tempered as we can only conclude on differences in ability of the viruses to infect human primary respiratory cells and not on differences in ability to replicate (i.e. produce new progeny viruses) in these cells. Accordingly, we modified the text in the corresponding result (lines 283, 318) and discussion (line 385) sections and the title of the manuscript to accommodate the reviewer’s comment.

2. Throughout the study, the authors use “URT” and “LRT” to differentiate between virus replication in different anatomical sites in the ferrets, however, this can be misleading and fails to capture some of the novelty of this work. In example, on lines 61-62 and 282-3 the authors state that this study is the first to show that virus replication in the URT of inoculated ferrets is the site of replication of transmitting influenza viruses. However, numerous other studies employing pre-pandemic H1N1 and seasonal H3N2 viruses (which typically are restricted to the URT for replication and do not replicate in lower respiratory tract tissues, including the lung) have already demonstrated a role for the URT in virus transmission. What makes this study unique (in this reviewer’s opinion) is identification of virus replication in nasal turbinates samples (in the potential absence of other tissues such as the trachea), and identification of potential cell types within the milieu of the nasal turbinates, in this property. It would likely be more precise to rephrase (in example, line 145, though there are several instances of this) that virus replication in nasal turbinates, and not “URT”, is associated with transmission. Furthermore, it is confusing to use “LRT” to define lower respiratory tract as not just the lung but also the entirety of the trachea – is an upper trachea sample truly representative of the “lower respiratory tract”? The authors were smart to collect multiple sections of both the trachea and lung in their studies; it is very unclear why they are not capitalizing on this and using more precise language throughout this study.

We thank the reviewer for this constructive suggestion about highlighting the novelty of our work. We define here the upper respiratory tract as the nasal turbinates, the paranasal sinuses, the pharynx and the larynx. The lower respiratory tract is taken to include the trachea, bronchus, lung (bronchioles and alveoli). This distinction between upper and lower respiratory tract is not based on a strict anatomical designation but on differences in physiology and mucociliary transport going downwards from the nasal turbinates and upwards from the terminal bronchioles (through bronchus/trachea) to the pharynx for clearance by entrance into the gastrointestinal tract (Handbook of Physiology, Section 3: Respiration. Volume 1. Chapter 8: Physiology of the upper airway, by D.F. Proctor. American Physiological Society, Washington D.C. Page 309). However, we acknowledge that

this is not a strict distinction and that the terminologies “upper” and “lower” respiratory tracts might have been used differently in previous articles in the field, creating confusion. For clarity, this definition was added to the figure legend of Figure S1 (additional figure added to answer the first minor comment). Moreover, to avoid any confusion about what tissue belong to the upper/lower respiratory tracts, we changed, throughout the text, the upper/lower respiratory tract designation to the name of the tissue, when possible. As an example, we described more precisely in the introduction the binding pattern of human and avian influenza A viruses to the respiratory tract line 55-60: “In humans, α 2,6-SA receptors are predominantly present on ciliated cells in the upper respiratory tract (URT), i.e. in the nasal turbinates, paranasal sinuses, pharynx and larynx, and in the lower respiratory tract (LRT), i.e. in the trachea and bronchus^{28,29}. In contrast, α 2,3-SA receptors are mainly present on bronchiolar non-ciliated cuboidal cells and alveolar type II pneumocytes of the LRT^{28,29}.” Based on this, we agree with the reviewer that human influenza viruses, through binding to α 2,6 sialic acids, target the trachea and to a certain extent bronchus (in the lower respiratory tract) and that our set of experiment allows to disprove the trachea (and bronchus) as the site of expulsion of airborne transmissible influenza A viruses. Text throughout the manuscript was adapted to highlight better this finding. However, as only the nasal turbinates were sampled as part of the upper respiratory tract in the transmission experiments, we feel that we cannot directly conclude that the airborne transmissible viruses were expelled from the nasal turbinates. We think that it is the combination of the two series of experiments (transmission and tissue tropism) that allowed us to conclude that the site of expulsion of airborne-transmissible viruses is probably the nasal respiratory epithelium.

3. The authors need to more clearly state how virus transmission is defined in this study. For H5N1-mut data shown in Figure 2 and supplemental Figure 3, the transmission data presented is not very convincing as presented. Recipient 15 does not appear to have any detectable infectious virus in nose or throat swabs (only nose swabs positive via PCR per supplemental data for this ferret, though Figure 3 indicates a T and not N, so this is unclear), and Recipient 13 has only one sample with positive infectious virus detection in the throat swab only; the nose is only positive via PCR. Specifically for Recipient 15, it seems very misleading to the reader to present data in Figure 3 highly suggesting virus detection in throat and nose swabs yet have this be via PCR only and not viral titers? This is not standard practice in the field. In this reviewer’s opinion, the H5N1 virus did not exhibit robust transmission in any pair, with only Recipient 13 possessing infectious virus in 1 of 2 collected samples, and the other 3 recipients not possessing infectious virus in nose or throat swabs collected at any time. Authors should state criteria for transmission definition in the text and temper their interpretation of transmission data from this component of the study accordingly.

In this study, we define transmission based on a CT-value threshold of 35. In the second set of transmission experiments (Figure 2), the detection of two consecutive swabs with a CT-value threshold of 35 was the criteria to euthanize the donor ferrets. Defining transmission based on CT-value allowed us to monitor in real-time whether transmission had occurred. Moreover, a CT-value threshold of 35 was the inclusion criteria for next-generation sequencing. We agree with the reviewer that this is not standard practice in the field, however the goal of this study was not to have a quantitative assessment of the kinetics and robustness of transmission but a qualitative assessment of which virus transmitted. We also agree with the reviewer that this was not well enough defined in the manuscript, and therefore we clarified this point in the figure legend of Figure 2. Additionally, we indeed observed that in the second set of transmission experiments (Figure 2) transmission was

delayed and less robust than observed in previous experiments with the same viruses, in which the donor ferret is only inoculated intranasally and stays in contact with the recipient ferret for fourteen days (rather than maximum 5 days). This was particularly the case for the A/H5N1_{AT} transmission and we tempered our interpretation of the transmission data accordingly lines 148-152: “Transmission was defined here by the detection of two consecutive swabs with a threshold value in RT-qPCR (Ct value) <35. Despite the fact that infectious virus titers were detected only in one throat swab of Recipient 13 (Figure S4), viral RNA was amplified from the other swabs of Recipient 13 and Recipient 15, allowing the characterization of the nature of the virus that had transmitted.” The reason for the absence of robust transmission might be the fact that the donor ferrets were suffering from lower respiratory tract disease, possibly leading to impaired breathing, or that the donor ferrets were removed from the experiment too early in some cases. We also added these points to the discussion lines 317-322. Finally, we thank the reviewer for seeing the mistake in Figure S3 (now Figure S4) in which the nose swabs instead of the throat swabs were marked as virus positive. This mistake was corrected.

4. The authors dedicate a substantial portion of the discussion towards the implications of this study in aerosol transmission and aerobiology, yet do not include any aerobiology data in this study. While inclusion of aerobiology-based text in the discussion is warranted, the absence of aerobiology data supporting this text makes the large focus in the discussion of this topic feel especially speculative, and again (in this reviewer’s opinion) diminishes the true novelty of this work.

We agree with the reviewer. Parts of the discussion on aerobiology were drastically shortened and more emphasis was put on the identification of the site of expulsion of airborne transmissible influenza A viruses.

Minor comments:

1. Collection of discrete nose and throat swabs are not frequently collected samples in this field (compared with nasal washings). It would be of benefit for the reader to include additional text in the methods or results regarding the anatomical positioning of these samples (especially the throat sample, and the relative proximity of this specimen in relation to the soft palate), so the reader can more readily interpret this data. Similarly, do the nose swabs collected encapsulate virus replication from both respiratory and olfactory epithelia? The authors discuss ex vivo ferret cultures inclusive of both cell types in the discussion (lines 340-345) and it is unclear to the reader how to contextualize infectious virus from nose swabs in this argument without a precise understanding of how this sample is collected.

We added a supplementary figure (Figure S1) showing schematically the anatomical positioning of the inoculation, swab collection and tissue collection. Although we cannot be certain about the origin of the viruses collected in the nose swabs, the fact that nose swabs collected from ferrets inoculated with viruses that replicate either in the respiratory or olfactory epithelium are virus positive suggests that the nose swabs encapsulate virus replication from both respiratory and olfactory epithelia.

2. Lines 96-7, please provide the percentage of virus detected in throat swabs following intratracheal inoculation similar to how this is presented in lines 101-102 – it is difficult to see visually (specifically for donor 3) in Figure 1.

The percentages of virus detected in the throat swabs upon intratracheal inoculation were indicated in the text line 102-104: “However, in the throat swabs of three donor ferrets, a low amount of the virus that was instilled in the LRT was also detected (Donor 1 (day 1

(17,8%), day 4 (76,5%), day 6 (22,2%), donor 3 (day 1 (2,4%)) and donor 4 (day 1 (94,6%), day 5 (9,9%) and day 6 (20,8%))”.

3. Specify in figure legends for Fig 1 and supplementary data that gray bars represent infectious virus.

Figure legends of Figure 1 and Figure S2, S3 and S4 were adapted accordingly to the comment of the reviewer.

4. Lines 116-7, “...evidence of active replication in the LRT was missing.” Is this statement referring to data from previous studies (in which case, provide a reference) or data not shown (in which case, specify this)? This statement seems to contradict with line 136 that states that virus instilled following intratracheal inoculation was indeed replicating.

We apologize for the confusion. This statement is referring to data from the first experiment of the present manuscript (Figure 1), in which only swabs were collected from donor ferrets. The virus that was instilled intratracheally was only detected in throat swabs of three donors out of four, and therefore, we did not have the proof that the virus that was instilled intratracheally was replicating in the lower respiratory tract. This prompt us to perform the second experiment of the present study, in which donor ferrets were sacrificed when transmission was observed or the latest at day 5 after inoculation to harvest parts of the upper and lower respiratory tract. Line 136 is referring to the second experiment, where the virus that was instilled intratracheally was indeed detected in the lower respiratory tract (trachea/lungs). The statement was modified to clarify our observations, lines 123-126.

5. Please provide a supporting reference for the statement on lines 259-61 regarding distance of particles expelled.

We added the following reference: Wells, W.F. On airborne infection: study II. Droplets and droplet nuclei. American Journal of Hygiene 20, 611-618 (1934).

6. Methods: please specify the order in which dual-virus ferret inoculations took place (did intranasal instillation occur before or after intratracheal instillation of virus)? Also specify the vendor of ferrets in this study.

The intratracheal inoculation was performed before the intranasal inoculation. The ferrets were purchased from Euroferret (Denmark) and TripleF (USA). This information was added to the Methods section.

REVIEWERS' COMMENTS:

Reviewer #2 (Remarks to the Author):

Authors have addressed all comments raised during initial peer review; no further comments.